# AMBIsome Therapy Induction OptimisatioN (AMBITION): High dose AmBisome for cryptococcal meningitis induction therapy in sub-Saharan Africa: economic evaluation protocol for a randomised controlled trial-based equivalence study

Ponego Lloyd Ponatshego,[1] David Stephen Lawrence,[1,2] Nabila Youssouf,[1,2] Sile F Molloy,[3] Melanie Alufandika,[4] Funeka Bango,[5] David R Boulware,[6,7] Chimwemwe Chawinga,[8] Eltas Dziwani,[4] Ebbie Gondwe,[4] Admire Hlupeni,[9] Mina C Hosseinipour,[8] Cecilia Kanyama,[8] David B Meya,[6] Mosepele Mosepele,[1,10] Charles Muthoga,[1] Conrad K Muzoora,[6] Henry Mwandumba,[4,11] Chiratidzo E Ndhlovu,[9] Radha Rajasingham,[7] Sumaya Sayed,[5] Shepherd Shamu,[9] Katlego Tsholo,[1] Lillian Tugume,[6] Darlisha Williams,[6,7] Hendramoorthy Maheswaran,[4,12] Tinevimbo Shiri,[11] Timothée Boyer-Chammard,[13] Angela Loyse,[3] Tao Chen,[11] Duolao Wang,[11] Olivier Lortholary,[13] David G Lalloo,[4,11] Graeme Meintjes,[5] Shabbar Jaffar,[11] Thomas S Harrison,[3] Joseph N Jarvis,[1,2] Louis Wilhelmus Niessen[11]

PLP and DSL contributed equally.

JNJ and LWN contributed equally.

For numbered affiliations see end of article.

**Correspondence to**
Dr David Stephen Lawrence;
david.s.lawrence@lshtm.ac.uk

## ABSTRACT

**Introduction** Cryptococcal meningitis is responsible for around 15% of all HIV-related deaths globally. Conventional treatment courses with amphotericin B require prolonged hospitalisation and are associated with multiple toxicities and poor outcomes. A phase II study has shown that a single high dose of liposomal amphotericin may be comparable to standard treatment. We propose a phase III clinical endpoint trial comparing single, high-dose liposomal amphotericin with the WHO recommended first-line treatment at six sites across five counties. An economic analysis is essential to support wide-scale implementation.

**Methods and analysis** Country-specific economic evaluation tools will be developed across the five country settings. Details of patient and household out-of-pocket expenses and any catastrophic healthcare expenditure incurred will be collected via interviews from trial patients. Health service patient costs and related household expenditure in both arms will be compared over the trial period in a probabilistic approach, using Monte Carlo bootstrapping methods. Costing information and number of life-years survived will be used as the input to a decision-analytic model to assess the cost-effectiveness of a single, high-dose liposomal amphotericin to the standard treatment. In addition, these results will be compared with a historical cohort from another clinical trial.

## Strengths and limitations of this study

► This economic analysis will provide evidence to inform policy decisions about the use of a more expensive medication in cryptococcal meningitis (CM).

► This analysis will provide data that may justify initiatives to increase the availability of a more expensive medication in CM.

► This approach will enable the development and application of country-level costing tools across five African country settings which can be reused for future studies and contribute to capacity building in the region.

► The study is taking place at six large referral hospitals across five countries in East and Southern Africa and the results might not be representative or generalisable to remote rural areas or settings in other countries.

**Ethics and dissemination** The AMBIsome Therapy Induction OptimisatioN (AMBITION) trial has been evaluated and approved by the London School of Hygiene and Tropical Medicine, University of Botswana, Malawi National Health Sciences, University of Cape Town, Mulago Hospital and Zimbabwe Medical Research Council research ethics committees. All participants will provide written informed consent or if lacking capacity will have consent provided

by a proxy. The findings of this economic analysis, part of the AMBITION trial, will be disseminated through peer-reviewed publications and at international and country-level policy meetings.

**Trial registration** ISRCTN 7250 9687; Pre-results.

## INTRODUCTION

Cryptococcal meningitis (CM) is a severe fungal infection of the brain which occurs in advanced HIV infection. It is estimated that there are roughly 220 000 cases of CM globally per year with 73% of these occurring in sub-Saharan Africa. Annual global deaths are estimated at 181 000 and CM is responsible for approximately 15% of all AIDS-related deaths.[1]

The current recommended first-line treatment for CM is amphotericin B deoxycholate (AmBd). AmBd is associated with multiple drug-induced toxicities including anaemia, impaired renal function, electrolyte abnormalities and infusion-related reactions which make it unsafe to administer in high doses. AmBd is also difficult to administer, requiring hospitalisation for 7 to 14 days of intravenous infusions, depending on which oral antifungal it has been paired with. In addition, treatment outcomes are poor with acute mortality at 10 weeks ranging from 30% to 55%.[2] The use of a liposomal form of amphotericin called Ambisome (hereafter referred to as L-AmB) is associated with reduced drug-induced toxicities when compared with conventional AmBd.[3] The long tissue half-life and effective penetration into the brain tissue of L-AmB has prompted research into the effectiveness of treatment with short courses of high-dose L-AmB.[4] The AMBisome Therapy Induction OptimisatioN (AMBITION) phase II clinical trial conducted in Botswana and Tanzania found that a single, high dose of 10 mg/kg L-AmB was well tolerated and led to a non-inferior reduction in fungal burden in cerebrospinal fluid when compared with standard 14-day courses of 3 mg/kg L- AmB.[5] This dosing strategy is now being taken to a clinical endpoint trial.

The phase III AMBITION trial is a phase III open-label randomised control non-inferiority trial to compare single, high-dose L-AmB treatment to the WHO first-line recommended regimen of a 7-day course of AmBd-based treatment in avoiding all-cause mortality in HIV-associated CM (figure 1).[6 7] Eligible patients will be randomised to receive either:

1. L-AmB 10 mg/kg day 1 given with 14 days of fluconazole 1200 mg/day and flucytosine 100 mg/kg/day (single dose) or
2. Amphotericin B deoxycholate 1 mg/kg/day for 7 days given with 7 days of flucytosine 100 mg/kg/day followed by 7 days of fluconazole 1200 mg/day (control arm).

After the 2-week induction phase, all patients will receive fluconazole 800 mg/day to 10 weeks and 200 mg/day thereafter. Antiretroviral therapy (ART) will be commenced 4 to 6 weeks after initiation of antifungal therapy. The trial will enrol 850 patients across six sites in five countries in Africa: Gaborone, Botswana (90); Blantyre (230) and Lilongwe (110), Malawi; Cape Town, South Africa (80); Kampala, Uganda (110) and Harare, Zimbabwe (230). All participants will be invited to take part in the economic evaluation study.

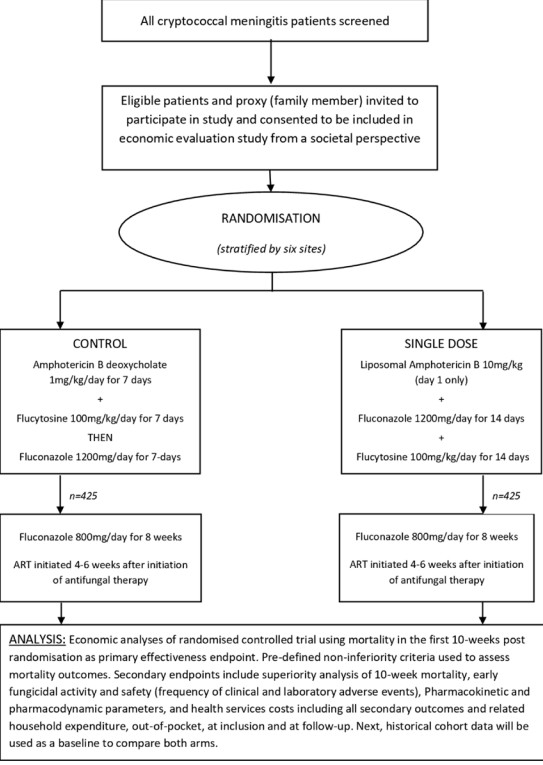

**Figure 1** Economic evaluation flow diagram: trial entry, randomisation, treatment and follow-up.

The use of L-AmB has potential implications for both clinical outcomes and healthcare costs. The widespread availability of L-AmB has previously been limited by the high cost of therapy: currently, the internationally listed price is $85 per 50 mg vial compared with $8 per 50 mg vial of AmBd. The listed cost per patient of the medication for the single-dose arm in this trial will be $996 versus $132 for the control arm. However, the impact of a potentially more clinically effective intervention that is associated with fewer drug-induced toxicities and a reduced length of hospital stay may offset this expense. An argument for widening access to L-AmB has been strengthened since, in September 2018, Gilead announced as part of the expanded access preferential pricing programme for visceral leishmaniasis to include CM. While the normal cost in other countries varies from US$80 to US$400, the drug will now be available for US$16.25 in 116 low/middle-income countries.[8] This could have a dramatic impact on mortality.[9]

We plan an economic analysis to estimate the cost consequences and the cost-effectiveness of short-course L-AmB treatment, compared with the control arm, in five individual country settings across sub-Saharan Africa. The findings will also be compared with a historical cohort from the recently completed Advancing Cryptococcal Meningitis Treatment for Africa (ACTA) trial. The ACTA trial recruited patients in Cameroon, Malawi, Tanzania and Zambia and compared treatment outcomes among individuals receiving one of five different treatment regimens, including the control arm used in the AMBITION trial.[10] The purpose of this comparison is to identify any change in costs over time and any gross variation in health service costs between the AMBITION and ACTA cohorts, and to enable comparison of the cost-effectiveness of the short-course L-AmB with the other regimens tested in ACTA.[11]

The hypotheses are that the short-course treatment:

1. Will show a zero-net societal cost change or that there will be societal cost savings from the short L-AmB treatment and an equivalent or increased effectiveness of the treatment reducing mortality over a patient lifetime.
2. Will be cost-effective in terms of life-years saved over a patient lifetime when compared with historical cohorts who received different combination treatment regimens in a recently completed clinical trial.

These analyses will aim to provide the economic evidence to support wide-scale implementation of short-course L-AmB treatment across sub-Saharan settings.

## OBJECTIVES

The main objective of the economic analysis is to assess the cost-effectiveness of single, high-dose L-AmB compared with the control arm treatment regimen for HIV-associated CM across the five country settings in six sites.

### Secondary objectives

► To assess the cost consequences from the societal and health service perspective of single, high-dose L-AmB compared with the control arm across the five country settings.
► To assess the total health service costs per patient at each country site.
► To assess out-of-pocket expenses incurred by patients and households at all trial sites.
► To assess the percentage of catastrophic household expenditure experienced by patients at each trial site.
► To compare the total societal costs per patient and cost-effectiveness of a single, high-dose L-AmB with historical cohorts that received different treatment regimens within the ACTA trial.

## METHODS AND ANALYSIS
### Study design

The study is a prospective economic evaluation from the societal perspective—including both health service and patient related perspectives—comparing the costs and effectiveness of the two interventions at each of the six trial sites across the five country settings. The two key components of this study are the collection of data concerning personal expenditure on health and the development of country costing tools, which will be applied to data concerning health service costs collected within the trial.

### Household expenditure

To estimate the societal costs at the patient and household level, they or their representatives will be interviewed at two points in time: within the first 5 days of randomisation and at their final face-to-face follow-up at week 10. The questionnaires, based on those used in the ACTA trial and further developed for this study, are designed to estimate their personal healthcare expenditure in the 4 weeks leading up to enrolment and during the trial. A summary of the questions included in the questionnaire are presented in box 1. In addition, the out-of-pocket healthcare expenditure by the individual and their household, loss of income incurred due to illness and loss of labour time of the patients themselves and their carers will be collected using methods adopted in trials of a similar nature.[12]

The interview questions will preferably be asked directly to the patient. If the patient is confused or has reduced consciousness due to CM, a relative or next-of-kin may provide proxy consent for them to enrol in the trial. It is unlikely that this person will be fully aware of the patient's financial situation, and in these cases, it may be necessary to wait for the patient to recover before asking them directly. In cases where patients have prolonged confusion or are felt to have a poor prognosis, these questions can be asked of the proxy. Data will be collected by study doctors and nurses and entered into the trial Electronic Data Capture (EDC) system: a uniform database to be used across all sites. An interview guide for those collecting data from patients

**Box 1** Structure of the health economics questionnaire for the AMBIsome Therapy Induction OptimisatioN study

- ► Personal health expenditure including on consultations, medication, travel time and costs
- ► Relative and/or household health expenditure in relation to the patient's condition
- ► The duration and severity of the illness episode
- ► Loss of productivity and time off work for both patient and relative/s
- ► Profession and educational attainment of patient
- ► Profession and educational attainment of the person who earns the highest income (if not the patient)
- ► Access to social security, welfare support and health insurance
- ► Household expenditure on food, utilities, rent and large purchases such as cars, furniture and electrical items to assess the socioeconomic status of the household
- ► The need for temporary loans or the sale of assets to fund healthcare and other costs in relation to the illness episode
- ► The level of disability and care needs of the patient

will ensure that nuances and country-specific idioms are acknowledged.

The above methods have been developed through an iterative process. Initially, the lead Health Economist, who led on the ACTA cost-effectiveness analysis, and members of the Trial Management Group developed the data collection tools and integrated these into the wider trial EDC. This was later refined following a 1-week meeting of health economists and study team members from across the AMBITION sites held in Blantyre, Malawi in November 2017. This provided the opportunity for experts working in this field and individuals who will collect data from patients to improve these tools by collecting and entering data from fabricated patients. Feedback was then integrated into both the data collection methods and the EDC and shared for final approval across the AMBITION consortium until a consensus was reached.

### Country-specific costing approaches

Presently, each site has differing levels of experience with conducting economic analyses and has varied access to validated country costing tools. As stated, the preparation of the trial included a 1-week workshop with at least two team members from each site to assess the face validity and completeness of the questionnaire, to practise electronic data entry of the completed questionnaire, carrying costing computations, and to increase the knowledge and understanding of economic evaluation. In each country, resource use data will be collected using an ingredients-based approach. The data on individual resource use will be collected from all participants onto case report forms. Overhead costs, including costs of admissions and laboratory tests, will be collated from the hospitals' financial and utilisation documents.

### Botswana Harvard AIDS Institute Partnership: Gaborone, Botswana

Botswana Harvard AIDS Institute Partnership will use a microcosting approach to estimate CM treatment in Botswana from a single health provider's perspective, in this case, from the Ministry of Health and Wellness perspective. The total of related costs, that is, patient-specific treatment cost and 'hotel costs' to cover CM treatment will be determined as per the 2016 Botswana HIV Treatment Guidelines.[13] All costs of pharmaceuticals will be taken from the listed tender prices at the Central Medical Stores, which procure stock and distribute pharmaceuticals and healthcare commodities to all government healthcare facilities. This package will provide an estimate of 'patient specific' costs of uncomplicated CM. 'Hotel costs' will determine the necessary hospital, staffing, capital and infrastructure requirements as the patient is admitted over a 7-day period. Data will be obtained from Princess Marina Hospital, which is the biggest, and main referral hospital in Botswana. Staff salaries will be taken from the Government of Botswana salary scales for health professionals. By combining this treatment costing data with the meningitis burden data generated through a previously completed audit, we will also generate an estimate of the total current costs to the Botswana health service of treating CM. The data will complement the 'Estimated resource needs for key health interventions offered under Botswana's Essential Health Services Plan (2013–2018)' that project the cost of all health programmes, including the treatment of CM, from 2013 to 2018 and will also be used as a reference for the next version of this document to be published in 2019.[14]

### Malawi Liverpool Wellcome Trust Clinical Research Centre: Blantyre, Malawi

Standardised national health costing data are not available in Malawi. An existing costing tool which was developed for an HIV testing study will be adapted.[15] Patient-related healthcare costs will be obtained from the Central Medical Stores Trust, the only supplier mandated to supply government health facilities in Malawi. Personnel costs will be refined by referencing the Malawi government salary structures and payroll and estimating the proportion of health personnel time taken in the clinical care of the patient, as well as allowance costs for patients working out of working hours. Programme-related costs will be adapted using the results from recent costing studies within the Queen Elizabeth Central Hospital.[15]

### University of North Carolina Project: Lilongwe, Malawi

Most of costing data for Lilongwe will be obtained using the same methods as that outlined above for Blantyre. In addition, local programme-related costs will be estimated and projected through a local costing study within the Kamuzu Central Hospital. This adaptation will be based on existing local costing data as part of the Driving Reduced AIDS-associated Meningo-encephalitis study, which will be shared with the AMBITION consortium.

## University of Cape Town: Cape Town, South Africa

There are currently no costing tools for primary data collection for disease-specific costing of CM treatment in South Africa. A costing tool will be developed from validated disease-specific costing tools for the South African context.[16] Costs from the health service (Department of Health) perspective will be collected using the ingredients costing approach and captured in an Excel spreadsheet. The quantity of resources used to treat CM in each study arm will be estimated from the trial and patient records. Prices for these treatment-related ingredients will be obtained from the various service providers within the Department of Health that are responsible for offering the products and services. Specific activities by staff will be identified and estimated through the review of routinely collected time sheets. Human resource costs as well as recurrent and capital costs will be allocated using hospital expenditure and financial records as well as records from the Provincial Department of Health. The average cost of each treatment component will be calculated by multiplying the quantity of resources used by the unit price. From this, we can calculate the cost per case of CM treated by multiplying the average cost by the number of times a particular cost has been incurred.

## Infectious Diseases Institute: Kampala, Uganda

Though there is a high burden of CM in Uganda, the costs associated with treatment have not been formally outlined in a costing tool. Previous research describing the costs of treatment used informally gathered estimates for treatment and management of the disease based on reports from various sources including local pharmacies, laboratories and the Ugandan Ministry of Health.[17] The Uganda team will create a systematic costing tool for CM. This tool will consider the costs of treatment as well as the costs borne by the patients being treated. The creation of this tool will involve input from the Ministry of Health as well as the major suppliers of medications. We will engage these organisations and other key stakeholders to ascertain the current costs of meningitis treatment and develop a costing tool which can be adjusted in the future should costs change.

## University of Zimbabwe School of Health Sciences: Harare, Zimbabwe

Clinical cost data will be collected alongside the clinical trial using microcosting methodologies. The economic evaluation will be done from the societal perspective to enable the study to assess the overall household economic impact of CM and will enable us to determine the patient and provider unit costs. This study is powered enough to detect both country-specific clinical and economic differences. Direct patient level clinical activity data such as drugs, staff time, diagnostics, pathology and radiology will be collected alongside the trial and relevant unit costs determined using the study protocol. In other cases, prices for drugs, diagnostic and radiology tests will be collected from the National Pharmaceutical Company of Zimbabwe and national reference diagnostic and radiology laboratories, and from consultations with experts. Indirect and overhead costs such as management and administration costs, utilities and other capital costs for in-patient days will be determined using data from Parirenyatwa Hospital, Harare financial records, the WHO Choice Database and from previous clinical trials that took place at the proposed site.[18 19]

## Data collection and data management

Data collected and validated using the EDC system will be stored in an electronic database that is protected using a scheme of authentication and encryption. Paper documents, such as clinical notes and administrative documentation, will be kept in a secure location and held for 5 years after the end of the trial. During this period, all data should be accessible to the competent or equivalent authorities, the sponsor and other relevant parties with suitable notice. Security of electronic records and data is a significant concern. All components of the distributed data systems will use authentication and encryption to render subject identity and personal health information unusable, unreadable or indecipherable to unauthorised individuals. Full Drive Encryption will be implemented at the hardware layer of all devices storing protected health information. A three-factor scheme will be used to authenticate users through the hardware layer to the application layer where personal health information is available. The applications will have user profiles to control access to certain data and reports. The application and database layers will use a combination of hashing and encryption for sensitive and personal data. Mobile devices and the staff operating them will not be equipped with the encryption keys to decrypt selected sensitive data fields.

## Confidentiality

The trial will be conducted in compliance with the approved protocol, the Declaration of Helsinki 2008, the principles of Good Clinical Practice and applicable national regulations. We plan to follow the principles of the UK Data Protection Act regardless of the countries where the trial is being conducted. Consent forms will be stored under the supervision of each local primary investigator in a secured office and accessible to trial staff only. The database will not hold personal details as participants are identified by their study number throughout the trial.

## Data analysis

As outlined above, information on resource use and number of units used will be collected at patient level through the EDC, as part of the trial, and through additional separate country costing studies. To validate and refine the data collection process, an early analysis will take place at each site after 10 patients have completed the 10-week study follow-up period.

On closure of the study, full data analysis will commence. First, an empirical cost-consequence analysis

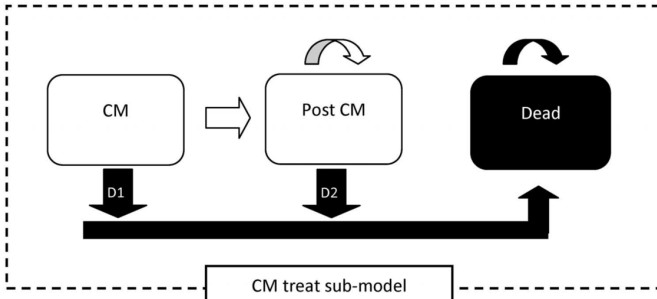

**Figure 2** Simplified Markov model structure to evaluate the cryptococcal meningitis (CM) treatment.[21]

will take place, using empirical individual patient data on societal resource use and unit cost based on the results from the costing studies.[20] Both societal and healthcare perspectives are chosen, and health service patient costs including household costs, treatment cost and hospitalisations in both arms will be compared over the trial period in a probabilistic approach, using Monte Carlo bootstrapping methods. To handle the heterogeneity of the trial population in within trial evaluation, we will derive a benefit value for each patient from the observed costs and effects and then construct a regression model with treatment variable and collected explanatory variables. Next, the costing information and number of life-years' survival will be used in a decision analytic model to assess the cost-effectiveness of the single, high-dose arm against the control arm and also against other combination treatment regimens from the ACTA trial. A Markov model has been chosen as it allows to explicitly account for passage of time to calculate time dependent costs and life-years of the remaining life span. Results will be presented using incremental cost-effectiveness ratios and cost-effectiveness acceptability curves generated by Monte Carlo bootstrapping methods. This approach avoids the stochastic fallacy and will determine if a single, high-dose L-AmB will be as or even more cost-effective compared with the current WHO recommended first-line treatment. An existing model will be adapted based on Jarvis *et al*, using the treatment submodel (figure 2).[21] The Markov model has a monthly cycle length, running the model for 12 cycles to calculate annual costs and annual life-years. We will supplement the model to be able to extrapolate data beyond the period of observed follow-up. The Markov modelling framework also allows for the synthesis of data from secondary sources, like mortality risk from other causes and excess mortality risks.[17] It also allows for probabilistic sensitivity analyses. We are going to use non-parametric bootstrapping to assess uncertainties, and both deterministic and probabilistic sensitivity analyses to examine the impact of all relevant parameters on the incremental cost-effectiveness ratio. The other country teams will be able to use an adapted model to enter their country-specific data and costing information as well as estimated survival figures, while using the pooled effectiveness information. In this way, each country will arrive at valid country-specific economic estimates.

We will follow the Consolidated Health Economic Evaluation Reporting Standards appraisal guidelines on economic evaluation.[22] As our study is an equivalence study using empirical data in a model-based analysis, these guidelines we will also include good modelling practice approaches.[23]

We anticipate the development of five different country-specific costing tools for CM which can be used to compose two five-country manuscripts concerning the cost consequence of CM across sites as well as the cost-effectiveness of the intervention across sites. In addition, individual country-level publications will be allowed, using the whole five-country trial database, complemented with local, more detailed country-level costing studies.

We intend to use these findings to provide an economic argument for the adoption of single, high-dose L-AmB in low/middle-income countries and to help influence guidelines and policy. Another important component of this study is capacity building across the African sites through the delivery of a health economics course and the ongoing mentoring of individuals and teams at each of the sites.

### Patient and public involvement

For the primary AMBITION clinical trial, a number of the sites have well-developed community groups and are experienced in engaging local communities when undertaking such studies. These groups were consulted prior to trial implementation, and they will be consulted regularly during trial conduct. To engage the wider community, we will work with community groups, HIV patient groups and local ministries of health to provide information about the trial, disseminate the results and to develop health education materials aimed at dispelling the current beliefs around meningitis, and encouraging early care-seeking.

### DISCUSSION AND CONCLUSION

This phase III clinical endpoint trial comparing single, high-dose liposomal amphotericin to the control arm treatment at six sites across five counties will provide valuable information on the comparative effectiveness with existing and other proposed strategies. This will be based on the effectiveness of simplified treatment strategies as well on the possibly increased safety. The proposed economic analysis of the equivalence trial for the Malawi situation will allow for a realistic comparison with settings where there is very limited coverage of appropriate treatment of CM in people with HIV. The estimates from other trial settings will help to document the generalisability of our findings. The economic information will be essential in the support of wide-scale implementation strategies and the formulation and testing of alternative delivery modes in all comparable sub-Saharan setting.

A clinically effective and safer treatment for CM in sub-Saharan Africa could have a dramatic impact on HIV-associated mortality in the region. This economic

analysis is essential to help justify any policy change towards increasing the availability of more expensive medication if it is proven to be cost-effective. This process will enable the development of country-specific costing tools across five African sites, which can be used for future studies and will build capacity in the region.

**Author affiliations**
[1]Botswana-Harvard AIDS Institute Partnership, Gaborone, Botswana
[2]Department of Clinical Research, London School of Hygiene and Tropical Medicine, London, UK
[3]Research Centre for Infection and Immunity, St. George's University of London, London, UK
[4]Malawi-Liverpool-Wellcome Trust Clinical Research Centre, Blantyre, Malawi
[5]Institute of Infectious Diseases and Molecular Medicine, University of Cape Town, Cape Town, South Africa
[6]Infectious Diseases Institute, Makerere University, Kampala, Uganda
[7]Department of Medicine, University of Minnesota, Minnesota, USA
[8]Lilongwe Medical Relief Trust (UNC Project), Lilongwe, Malawi
[9]Department of Medicine, University of Zimbabwe, Harare, Zimbabwe
[10]Department of Internal Medicine, University of Botswana, Gaborone, Botswana
[11]Department of Clinical Sciences and International Public Health, Liverpool School of Tropical Medicine, Liverpool, UK
[12]Population Evidence and Technologies, University of Warwick, Coventry, UK
[13]Molecular Mycology Unit and National Reference Centre for Invasive Mycoses, Institut Pasteur, Paris, France

**Contributors** PLP and DSL jointly wrote the manuscript with contribution on the methodology from TS, LWN, TSH and JNJ. DSL is the international lead clinician for the trial and created the initial data collection tools for this sub-study. NY is the trial manager and SFM is the trial epidemiologist and both have provided critical input into the data collection tools, electronic data capture system and this manuscript. DRB, MCH, CK, DBM, MM, CKM, HM and CN are principal investigators at each of the sites and will oversee the development of costing tools and the collection of data at each site. MA, FB, CC, ED, EG, AH, CM, RR, SSa, SSh, LT, KT and DW are all study team members or affiliated academics who attended the Ambition health economics course and are the focal individuals for the health economics study at their respective sites, as well as contributing to the refining of the data collection tools and the country-specific sections of this manuscript. TC and DW are statisticians for the study and have contributed to the database function of the electronic data capture system. TS and HM are health economists who have contributed to the development of the methodology and TS will perform the early reviews of data and the overall data analysis with DSL. TBC is an international clinical adviser and a monitor who performs data checks for the study. AL, OL, DGL, GM and SJ provided expert input into the conceptualisation and design of both the broader study and this health economics study, particularly in terms of adopting a societal perspective. TSH and JNJ conceived and designed the broader study and are the coprincipal investigators. LWN leads the health economics study and has provided oversight of the entire process. All authors read, critiqued and approved the final manuscript.

**Funding** The study is jointly funded through the European & Developing Countries Clinical Trials Partnership (EDCTP), the Swedish International Development Cooperation Agency (SIDA) and the Wellcome Trust/Medical Research Council (UK)/UKAID Joint Global Health Trials.

**Competing interests** JNJ and TSH were the recipients of a Gilead Investigator Initiated Award (completed). TSH has received speaker fees from Gilead Sciences and Pfizer.

**Patient consent for publication** Not required.

**Ethics approval** The Research Ethics Committee of the London School of Hygiene and Tropical Medicine have approved the AMBITION trial protocol v2.1 07.11.17 which outlines this economic analysis (Ref 14355). Approval has also been granted by the following: University of Botswana Office of Research and Development (UBR/RES/IRB/BIO/042), Botswana Ministry of Health and Wellness Health Research and Development Division (HPDME: 13/18/1), Princess Marina Hospital Research and Ethics Committee (PMH 5/79(407-1-2017), University of Cape Town Human Research Ethics Committee (642/2017), Malawi National Health Sciences Research Committee (1907), Mulago Hospital Research and Ethics Committee (MHREC 1297) and the Medical Research Council of Zimbabwe (MRCZ/A/2263). Any amendments

will be submitted and approved by each ethics committee. All participants will provide written informed consent or if lacking capacity will have consent provided by a proxy. The findings of this economic analysis, which is embedded into the AMBITION trial, will be disseminated through peer-reviewed publications and at international and country-level policy meetings.

**Provenance and peer review** Not commissioned; externally peer reviewed.

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
