## [Reviewer comments · BMJ Open]

ARTICLE DETAILS

TITLE (PROVISIONAL)	AMBIsome Therapy Induction Optimisation (AMBITION): High dose Ambisome for Cryptococcal Meningitis Induction Therapy in sub-Saharan Africa: Economic Evaluation Protocol for a Randomised Controlled Trial Based Equivalence Study
AUTHORS	Ponatshego, Ponego; Lawrence, David; Youssouf, Nabila; Molloy, Sile; Alufandika, melanie; Bango, Funeka; Boulware, David R.; Chawinga, Chimwemwe; Dziwani, Eltas; Gondwe, Ebbie; Hlupeni, Admire; Hosseinipour, Mina C.; Kanyama, Cecilia; Meya, David; Mosepele, Mosepele; Muthoga, Charles; Muzoora, Conrad; Mwandumba, Henry; Ndhlovu, Chiratidzo; Rajasingham, Radha; sayed, sumaya; Shamu, Shepherd; Tsholo, Katlego; Tugume, Lillian; Williams, Darlisha; Maheswaran, Hendramoorthy; Shiri, Tinevimbo; BOYER-CHAMMARD, Timothée; Loyse, Angela; Chen, T; Wang, Duolao; Laloo, David; Meintjes, Graeme; Jaffar, Shabbar; Harrison, Thomas; Jarvis, Joseph; Niessen, Louis

VERSION 1 - REVIEW

REVIEWER	Amdres Henao University of Colorado Denver USA
REVIEW RETURNED	31-Oct-2018

GENERAL COMMENTS	Authors describe an economic evaluation protocol analysis to assess the cost-effectiveness of a single high dose of Amphotericin B vs standard therapy for Cryptococcal meningitis. This is an important follow-up study required after completion of the phase III AMBITION trial. - I'd clarify why not use a high single dose of Amphotericin B deoxycholate instead of the liposomal Amphotericin B in the treatment group.
--

REVIEWER	Claudia Frola Hospital General de Agudos "Dr. Juan A. Fernández" Fundación Huésped Buenos Aires, Argentina
REVIEW RETURNED	11-Nov-2018

GENERAL COMMENTS	Household expenditure "The questionnaires, based on those used in the ACTA trial and further developed for this study..."
--

	-The questionnaires used in the cited ACTA study are not available. Data analysis "To validate and refine the data collection process an interim analysis will take place after the first 20 patients have been recruited" -It would be the first 20 patients for each sites? This would ensure a more homogeneous validation in data collection at all of them.
--	--

REVIEWER	Fernando Fernandez-Llimos University of Lisbon, Portugal
REVIEW RETURNED	10-Dec-2018

GENERAL COMMENTS	This manuscript presents a protocol for an economic evaluation of a phase III randomised controlled trial (AMBITION - AMBIsome Therapy Induction OptimisatioN) of high doses of liposomal amphotericin B for cryptococcal meningitis in sub-Saharan Africa. This protocol is interesting, scientifically sound and well-written. However, I have a couple of minor concerns about the methods which the author might want to consider:  1) In the introduction authors should also incorporate information on other amphotericin B formulations (e.g. infused with intralipid, amphotericin B lipid complex, amphotericin B colloidal dispersion). Please, better justify the model based only on liposomal amphotericin B. 2) The articles "Efficacy and safety of amphotericin B formulations: a network meta-analysis and a multicriteria decision analysis" (J Pharm Pharmacol. 2017 Dec;69(12):1672-1683) and "Cost-effectiveness of amphotericin B formulations in the treatment of systemic fungal infections" (Mycoses. 2018 Oct;61(10):754-763) could be mentioned. 3) The micro-costing methodologies should be clearly described in methods section. 4) Please better specify the reasons for choosing Marvok model rather than a simple model (e.g. decision tree) for this disease. How the disease' states were defined? 5) Are you not planning to perform analyses on quality of life as well? Why? 6) Better describe the sensitivity analyses that you aim to perform in the model.
--

VERSION 1 – AUTHOR RESPONSE

'I'd clarify why not use a high single dose of Amphotericin B deoxycholate instead of the liposomal Amphotericin B in the treatment group'

Amphotericin B deoxycholate is a toxic and difficult to administer drug. At the current maximal therapeutic dosing of 1 mg/kg, it causes electrolyte imbalances, nephrotoxicity and anaemia to patients. At the site of administration, it causes drug-related infusion reactions and thrombophlebitis. Using a single, high-dose of the drug is not possible as it could precipitate a potentially fatal infusion

reaction and cause significant long-term toxicity. We have added a sentence to this effect in the manuscript.

'The questionnaires, based on those used in the ACTA trial and further developed for this study... The questionnaires used in the cited ACTA study are not available.'

The full questionnaires are too lengthy to be included in the manuscript thus are not provided but we have given a summary of the questions included in the health economics questionnaire in Table 1 and added a reference to an ACTA costing poster.

'To validate and refine the data collection process an interim analysis will take place after the first 20 patients have been recruited" - It would be the first 20 patients for each sites? This would ensure a more homogeneous validation in data collection at all of them.'

Thank you for raising this point. We have reviewed the proposed plan to perform an interim analysis after the recruitment of the first 20 patients across the entire consortium and have decided that an early analysis will now take place at each site after ten patients have completed the 10-week study follow-up period.

'In the introduction authors should also incorporate information on other amphotericin B formulations (e.g. infused with intralipid, amphotericin B lipid complex, amphotericin B colloidal dispersion). Please, better justify the model based only on liposomal amphotericin B.'

Although these other formulations do exist, liposomal amphotericin B is the only alternative formulation of amphotericin B that is widely produced, distributed and has been extensively trialled in the context of HIV-associated cryptococcal meningitis. It will now also be more affordable after Gilead, in September 2018, announced a reduction in the price of LAB for cryptococcal meningitis in HIV. We have added a sentence to this effect in the manuscript.

'The articles "Efficacy and safety of amphotericin B formulations: a network meta-analysis and a multicriteria decision analysis" (J Pharm Pharmacol. 2017 Dec;69(12):1672-1683) and "Cost-effectiveness of amphotericin B formulations in the treatment of systemic fungal infections" (Mycoses. 2018 Oct;61(10):754-763) could be mentioned.'

In their work, Tonin et al (2017) compare all available formulations of amphotericin: conventional Amphotericin B; Amphotericin B lipid complex (ABLC); colloidal dispersion or Amphotericin B colloidal dispersion (ABCD); liposomal Amphotericin B (LAB) and Amphotericin B in Intralipid. As discussed, liposomal amphotericin B is the only alternative formulation of amphotericin B that is widely produced, distributed and has been extensively trialled in the context of HIV-associated cryptococcal meningitis. In our opinion it would be confusing to introduce these different formulations into the discussion when we have already outlined in detail the superior safety profile of liposomal amphotericin when compared to conventional amphotericin B.

In the paper by Borba et al (2018) a modelling analysis is performed in the context of the treatment of all invasive fungal infections in Brazil, not specifically HIV-associated cryptococcal meningitis, and using exceptionally different durations of therapy (up to six weeks) and listed costs for each medication. The AMBITION cost-effectiveness analysis is unique in that we are prospectively comparing a single, high-dose of a more expensive drug with standard seven-day treatment with a less expensive medication and assessing if we can offset the drug costs with the savings of reduced hospital admissions and drug-related toxicity.

'The micro-costing methodologies should be clearly described in methods section.'

We have added the following sentences in the methods section:

In each country, resource use data will be collected using an ingredients-based approach. The data on individual resource use will be collected from all participants onto case-report forms. Overhead costs, including costs of admissions and laboratory tests, will be collated from the hospitals' financial and utilisation documents.

'Please better specify the reasons for choosing Markov model rather than a simple model (e.g. decision tree) for this disease.'

A Markov model has been chosen as it allows to explicitly account for passage of time to calculate time dependent costs and life years of the remaining lifespan. It functions better for computing long-term survival outcomes.

'How the disease states were defined?'

Disease states will be derived from a previous model that was developed by our group, based on the occurrence of the clinical disease stages.

'Are you not planning to perform analyses on quality of life as well? Why?'

We are not planning to perform analyses on quality of life. The study duration is 10 weeks, which is a short period of time to measure significant differences in quality of life. The existing data also suggest that despite a high acute mortality, survivors tend to recover without severe long-term disability. We are therefore using life years gained as the primary effectiveness outcomes.

'Better describe the sensitivity analyses that you aim to perform in the model.'

We have added the following sentence

We are going to use non-parametric bootstrapping to assess uncertainties, and both deterministic and probabilistic sensitivity analyses to examine the impact of all relevant parameters on the incremental cost-effectiveness ratio.

'Please provide another copy of your figures with better qualities'

We have uploaded higher quality photos with this resubmission.

'Patient and Public Involvement'

We have added the relevant section to the revised manuscript.

Thank you again for reviewing this manuscript. We look forward to receiving your feedback on this revised version.

VERSION 2 – REVIEW

REVIEWER	Andres Henao University of Colorado Denver, USA
REVIEW RETURNED	31-Jan-2019

GENERAL COMMENTS	Accept
--------

REVIEWER	Claudia Frola Hospital "Juan A. Fernández" Fundación Huésped Buenos Aires, Argentina
REVIEW RETURNED	10-Feb-2019

GENERAL COMMENTS	The suggestions made in the first review have been incorporated correctly.
--

REVIEWER	Fernando Fernandez-Llimos University of Lisbon
REVIEW RETURNED	05-Feb-2019

GENERAL COMMENTS	Thank you for the opportunity to review this new version. The authors attended to all my comments. No further comments are required.
--